# FUSING 3D-CNN AND LIGHTWEIGHT SWIN TRANSFORMER NETWORKS FOR HSI

**Baisen Liu**[1,2]**, Yuanjia Liu**[1,*]**, Wulin Zhang**[1]**, Yiran Tian**[1]

[1] Mudanjiang Normal University, Mudanjiang 157011, China
[2] Heilongjiang Institute of Technology, Harbin 150001, China
[*] `1023321561@stu.mdjnu.edu.cn`

## ABSTRACT

Recently deep learning has occupied an important position in hyperspectral image (HSI) classification. In this study, we explore the advantages of using convolutional neural networks (CNN) for feature extraction and fusing an advanced shift-window (swin) transformer network based on the transformer model for HSI classification. The swin transformer network attention perception, capable of learning local and global features, can avoid the dependence on single features during HSI classification. The experiments show that our proposed model outperforms traditional machine learning models, and achieves competitive results with advanced models. The source code can be found at https://github.com/MinatoRyu007/CNN-Swin.

## 1 INTRODUCTION

Hyperspectral image classification are remote sensing images that contain both image information and spectral information. The CNN have been one of the hottest research hotspots in pattern recognition for a long time, and the HSI classification field is no exception (Lee & Kwon (2017)). Swin Transformer is a Vision Transformer (VIT) network based on the self-attentive mechanism (Liu et al. (2021)). The transformer-based architecture processes patch through fixed-length contexts and are limited when applied to HSI classification tasks (He et al. (2021)). It is worth thinking about using CNN to replace the part of swin transformer model originally used for preprocessing and feature extraction. Therefore, this work proposes the combination of CNN and swin transformer classifier to improve HSI classification accuracy.

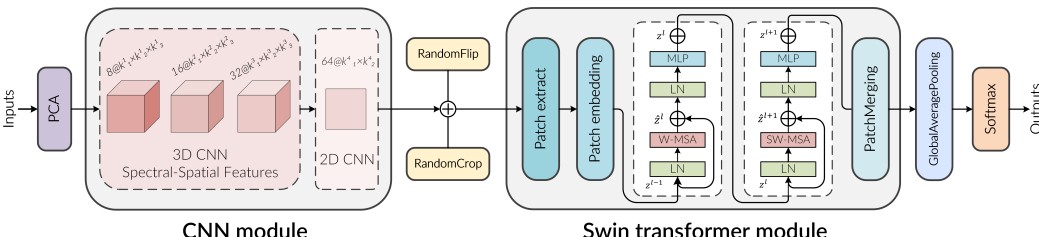

Figure 1: The proposed classification model fusing CNN and Swin transformer structure.

## 2 PROPOSED APPROACH

While the visual transformer model is making a splash in various fields of pattern recognition, we realize its potential in the field of HSI classification. However, the visual transformer model alone cannot handle the complex and high-dimensional HSI features well, so we consider combining it with the excellent feature extraction capability of CNN. We consider combining it with the excellent feature extraction ability of CNN and integrating it into the Swin transformer structure with global

self-attentive computation, which can also effectively complement the limited perceptual field of the CNN classifier.

Fig 1 shows the structural diagram of the model proposed in this work. As shown in the figure, the first part is a feature extraction CNN module consisting of three 3D-CNN layer and one 2D-CNN layer(Roy et al. (2019)). PCA is a dimensionality reduction for HSI. The function of the 3D-CNN module is to extract spatial-spectral features. The 2D-CNN module processes it to conform to the dimensions of the swin transformer module. Each layer uses the Relu activation function. The random-crop and random-flip are the most commonly used data augmentation method in DL. The Patch Embedding part is to crop the feature data to the settings window size and embed it in each patch. The function of the Patch Merging module is to subsample before the beginning of each Stage. Finally, the global average pooling layer will get the output with softmax activation function corresponding to different classes.

Two successive swin transformer blocks are the core of the swin transformer (1). The features $z^{l-1}$ input to this stage go through the Layer Normalization (LN), Multi-head Self-Attention (W-MSA), and residual layer in turn to get $\hat{z}^l$. After going through LN and Multilayer Perceptron (MLP) again, it enters the second block that has Shifted Windows Multi-head Self-Attention (SW-MSA).

$$\hat{z}^l = W - MSA(LN(z^{l-1})) + MSA(LN(Z^{l-1})) + z^{l-1}$$
$$z^{l+1} = MLP(LN(\hat{z}^{l+1})) + \hat{z}^{l+1} \qquad z^l = MLP(LN(\hat{z}^l)) + \hat{z}^l \tag{1}$$
$$\hat{z}^{l+1} = SW - MSA(LN(z^l)) + MSA(LN(Z^l)) + Z^l$$

## 3 EXPERIMENTAL

We use the Indian Pine (IP) dataset, the Salinas (SA) and the Pavia University (PU) datasets. In this study, The model uses Adam optimizer, Tensorflow 2.6 and python 3.8. Convolution kernel is $3 \times 3 \times i$. In the swin transformer model, we use a patch size of 2×2, a dropout rate of 0.03, a number of attention heads of 8, an embedding dimension of 64, a number of multilayer perceptrons of 256, a window size of 2 and shift window step of 1. The model under comparison uses open source projects as its source of code. This tool contains all the models involved in this experiment for comparison, including SVM and Baseline (Audebert et al. (2019)). The hyperparameters of the comparison test are set as follows: epochs=40, learning rate=0.01, training sample=30%.

Table 1 shows the comparison of the metrics of the models under the average accuracy (AA), overall accuracy (OA) and Kappa evaluation systems. It can be seen that the proposed model has the same excellent performance as the current mainstream models in terms of both classification reality map and accuracy, and is better than the traditional machine learning models (Hu et al. (2015))(Hamida et al. (2018)) (Lee & Kwon (2016))( Li et al. (2017)). See the appendix for a richer comparison.

Table 1: Comparison results of the classification in the three Public datasets (%).

| Methods | Indian Pines | | | Salinas | | | PaviaU | | |
|---|---|---|---|---|---|---|---|---|---|
| | AA | OA | Kappa | AA | OA | Kappa | AA | OA | Kappa |
| SVM | 0.317 | 0.528 | 0.427 | 0.317 | 0.528 | 0.427 | 0.635 | 0.834 | 0.771 |
| baseline | 0.654 | 0.753 | 0.713 | 0.908 | 0.917 | 0.908 | 0.852 | 0.961 | 0.948 |
| 1D CNN | 0.203 | 0.449 | 0.324 | 0.809 | 0.832 | 0.812 | 0.585 | 0.781 | 0.696 |
| 3D CNN | 0.663 | 0.775 | 0.737 | 0.879 | 0.911 | 0.901 | 0.857 | 0.954 | 0.939 |
| 3D FCN | 0.545 | 0.702 | 0.658 | 0.904 | 0.939 | 0.932 | 0.867 | 0.976 | 0.968 |
| S-S 3DCNN | 0.744 | 0.779 | 0.749 | 0.908 | 0.947 | 0.941 | 0.876 | 0.969 | 0.959 |
| Ours | 0.988 | 0.997 | 0.9974 | 0.998 | 0.999 | 0.999 | 0.996 | 0.998 | 0.998 |

## 4 DISCUSSION

In this paper, a new network with deep CNN and state-of-the-art swin transformer fusion is proposed. The excellent feature extraction capability of CNN is used to extract complex spatial spectral joint features in hyperspectrum. And these features are sent to the lightweight swin transformer global attention container for training.

ACKNOWLEDGEMENTS

The work was supported by the Natural Science Foundation of Heilongjiang Province for Key projects, China (Grant no. ZD2021F004), the Postdoctoral Scientific Research Developmental Fund of Heilongjiang Province, China (Grant no. LBH-Q18110)

URM STATEMENT

Author Yuanjia Liu meets the URM criteria of ICLR 2023 Tiny Papers Track.

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

## 5 APPENDIX

Figure2, 3 and 4 shows the classification chart of the IP dataset for the classification results in our proposed model. Our proposed model achieves the best visual performance on the classification graph. Not only does it perform well on the IP dataset, but also on the SA and PU datasets.

Figure 5 shows the accuracy function and loss function in our model.Figure 6 depicts a graphical comparison of the classification accuracy of various models in the IP, SA and PU datasets as training progress increases. In comparison to other models, our model achieves higher accuracy in shorter training epochs and has a more stable accuracy curve.

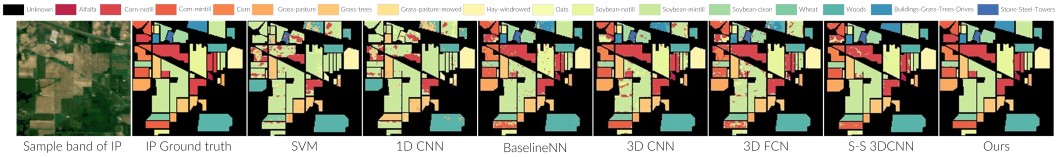

Figure 2: Comparison of IP dataset classification chart with other models.

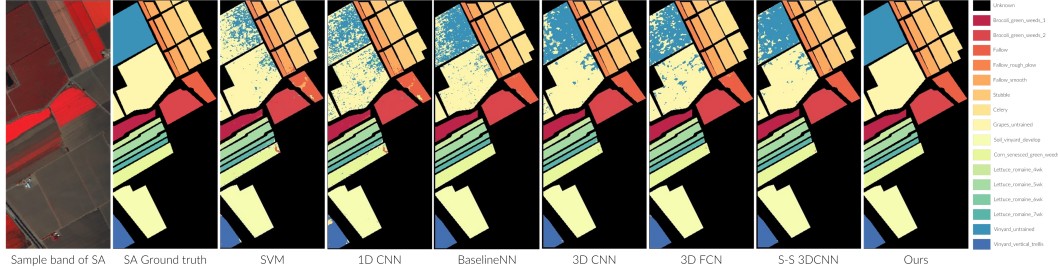

Figure 3: Comparison of SA dataset classification chart with other models.

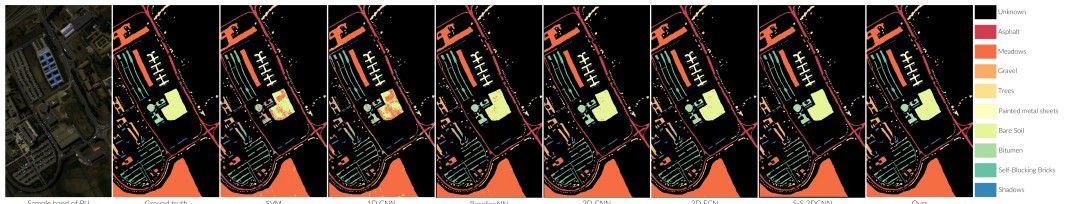

Figure 4: Comparison of PU dataset classification chart with other models.

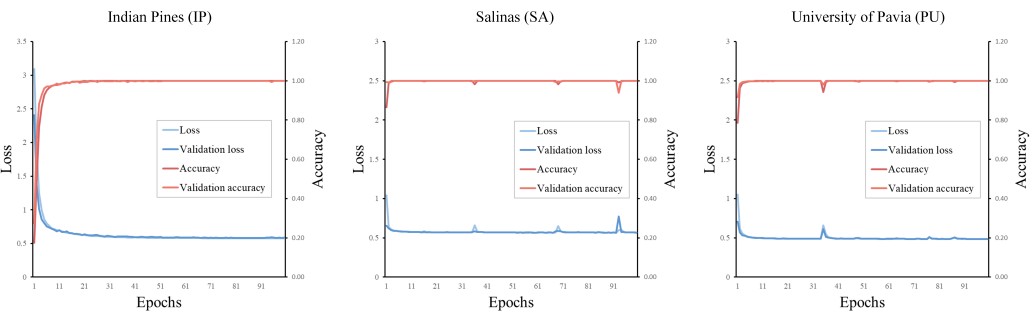

Figure 5: The validation accuracy and validation loss of our model in IP datasets, SA datasets and PU datasets.

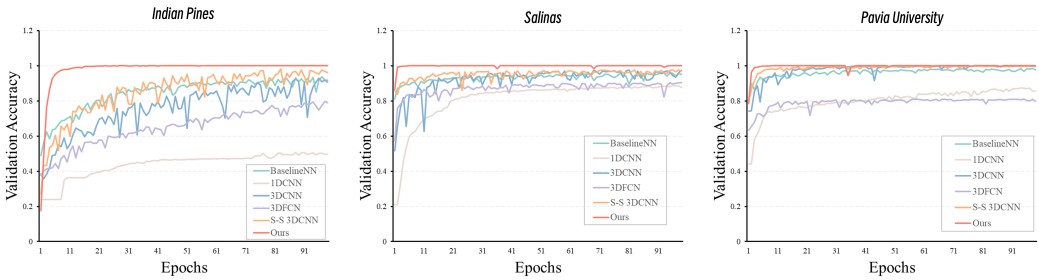

Figure 6: Comparison of model validation accuracy in the IP, SA and PU datasets.

