# OpenReview forum: "Fusing 3D-CNN and lightweight Swin Transformer networks for HSI"
_ICLR.cc/2023/TinyPapers — Submitted to Tiny Papers @ ICLR 2023_

### Official Review · Reviewer_VjZT · 2023-03-26

**Confidence:** 5

**Summary Of Contributions:**

a network structure using CNN and SwinTransfomer is proposed for hyperspectral image classification

**Rating:**

Needs Clarification (NC): a submission which does not meet the reviewing criteria and needs clarification for its described problem or solution

**Strengths And Weaknesses:**

S1: The performance of the proposed structure is good.

W1: It is not clear to me the motivation of such a new structure design, section 2 only gives the design of the model, but did not give any explanation on it.

W2: When comparing the model, it is better to include the number of parameters of each model, the experiments in the paper does not give this information, so it is hard to say if the comparison is fair.

W3: The paper is not anonymized, the name of the author is in the URM statement.

**Suggested Changes:**

C1: I would like the author to clarify the motivation behind the model design.

C2: It would be a stronger submission if the experiments can show more details such as the number of parameters in each of the models.

---

> ### Author Response · Authors · 2023-05-17
> **Revised version with added motivation and experimental details**
>
> Comments from the reviewers are greatly appreciated. In the revised version, we added the motivation and experimental parameters for using this model.
> 1. W1:CNN has shown excellent performance as a general backbone network in the field of computer vision, while the Transformer architecture from the NLP field has brought about a new systemic transformation. Combining these two structures will inevitably lead to higher classification efficiency. We believe that using CNN to extract features and combining them with visual Transformer classification is an efficient architecture.
>
> 2. W2:In the comparative experiment section, we used open source models. Hyperparameters such as learning rate, epochs, and the ratio of training to test samples were set to be the same. More detailed parameters, such as convolution kernel size and activation function, were not set the same because they can affect the stability of the architecture. However, the parameter settings and experimental source code address for our model will be made public in the revised version.
>
> 3.  W3: We apologize for the inclusion of author names in the review version due to our lack of experience with initial submissions.

---

### Official Review · Reviewer_WG1Q · 2023-03-30

**Confidence:** 3

**Summary Of Contributions:**

The paper presents a neural architecture that fuses 3D CNN and Swin Transformer for the task of hyperspectral image (HSI) classification. The model is tested on 3 HSI datasets and provide quantitative and qualitative results.

**Rating:**

Great Start (GS): a submission which meets some of the reviewing criteria but has room for improvement

**Strengths And Weaknesses:**

While the used architectures are not novel, their combined use for HSI classification is novel. The reported results are far better than the reported competitors and the qualitative samples are almost identical to the ground truth.

However, the paper has room for improvement.

First of all, the description of the model is not clear to me and I think important details are missing to be reproducible:
- it is not described how PCA is used to pre-process the input;
- it is not clear what the output of the model is. From the text and Fig. 1, I would have expected a single classification for image, given the global average pooling, while the model seems to predict per-pixel classification looking at Figures 2, 3, 4;
- the loss function used for training is not reported;
- Equation 1 contains several symbols that are not defined. What are W, SW, LN, Z, z, etc.?
- Fig. 1 seems to contain an error: the original swin transformer model contains a regular-window W-MSA module followed by a shifted-window SW-MSA module while the figure in the paper shows two W-MSA modules;

In addition, in the experimental results section, there are some details that should be clarified:
- What does it mean to have a 70% test set ratio? Is it the test set 30% of the data?
- Was a validation set used to define the hyperparameters?
- Was the train/val/test split the same one used by other methods too?
- Why aren't all the models shown in Fig. 2, 3, 4 reported also in Table 1?
- Why aren't the methods presented by Roy et al. (2019) in Table V reported in Table 1 for comparison? In particular, the approach proposed by Roy et al. (2019) achieves comparable accuracy with the proposed method.

**Suggested Changes:**

I encourage the authors to address the points reported above by:
- improving the description of the proposed method and adding the requested details, in particular the output format and the loss function;
- clarifying the data used for training/validation/testing;
- updating the comparison with the literature to have a fair comparison. The paper should include the methods presented in previous work (e.g. those in Roy et al. (2019)) or explain why it is not possible to compare with those methods.

The revised version of the paper could also update the text to remove some issues with the English language and the format, e.g.:
- the abstract contains "The swin transformer network attention perception, " that seems a typo;
- the intro contains: "self-attentive mechanism" which is usually referred to as "self-attention mechanism", "setting.This", "data, and Compress";
- the paper uses the term "in this letter" instead of using "in this paper" or "in this work";
- "training calendar time" meaning "training progress".

---

> ### Author Response · Authors · 2023-05-17
> **The revised version corrects errors in syntax and structure diagrams, and adds some experimental details**
>
> Comments from the reviewers are greatly appreciated.
> 1. We used CategoricalCrossentropy as the loss function. We are prepared to make the experimental source code publicly available so that later researchers can study the specific parameters of the model.
>
> 2. We have corrected the framework diagram and terminology, and supplemented the parameters for comparative experiments. We have also enriched the comparative data, and the model used for experimental comparison is from an open source model that has been cited in our reference.

---

### Author Response · Authors · 2023-04-11
**A brief description of the changes in the revised version**

We have updated the manuscript based on the corrections recommended by the reviewers.
1, Our experiments' source code is made public in the abstract.
2, Include our justification for using fusion models in Chapter 2.
3, The experimental hyperparameters updated for the studies are compared in Chapter 3.
4, Graphs of loss functions have been included in the appendix.
5, Some of the grammatical errors, unexplained terms, and errors in structural diagrams raised by the reviewers were corrected.

---

### Meta-Review · Area_Chair_hdS6 · 2023-04-07

**Recommendation:** Invite to archive
**Confidence:** 4

**Metareview:**

Both the reviewers agree that the paper lacks clear motivation for infusing 3DCNN with the Swin transformer and misses important implementation details such as
- how the hyperparameters of the infused model are chosen,
- what is the loss function used to train the model,
- how the model fairly compares against existing models, and
- what are the issues with the existing model that this paper is trying to solve?

I think this is a great start and authors are encouraged to clearly explain the motivation and provide enough details about the proposed architecture.

**Summary:**

The paper's main message is to infuse two well-studied neural architectures 3D CNN and Swin Transformer for the task of HSI classification. Though this infusion is considered novel by the reviewers, the authors do not fairly motivate the need for such incorporation.

**Reason For Not Giving A Higher Recommendation:**

As one of the reviewers noted that the proposed infusion of architecture is novel and observed exciting results. I recommend revising the submission from two aspects:
- Clearly state the motivation for this infusion. Think about what benefits these two models provided independently and what pros of each model you want to use.
- Provide enough details about the experimental settings, so that the readers of the work reproduce results.

**Reason For Not Giving A Lower Recommendation:**

NA

---

### Decision · Program_Chairs · 2023-04-10

Invite to archive